# Measurement Invariance of the Self-Stigma of Mental Illness Scale: A Cross-Cultural Study

**DOI:** 10.3390/ijerph19042344

**Published:** 2022-02-18

**Authors:** Shengnan Li, Patrick J. Heath, Carlos A. Vidales, David L. Vogel, Yangang Nie

**Affiliations:** 1Department of Psychology, School of Education, Guangzhou University, Guangzhou 510006, China; lishengnan010@gmail.com; 2Department of Psychological Science, Gustavus Adolphus College, Saint Peter, MN 56082, USA; pheath@gustavus.edu; 3Department of Psychology, College of Liberal Arts and Sciences, Iowa State University, Ames, IA 50011, USA; cvidales@iastate.edu (C.A.V.); dvogel@iastate.edu (D.L.V.)

**Keywords:** measurement invariance, cross-cultural, self-stigma, mental illness

## Abstract

The current study assessed the measurement invariance of the Self-stigma of Mental Illness scale (SSOMI) across Chinese and US samples and assessed whether the SSOMI differentially relates to distress levels across Chinese and US participants. We included 487 participants in China and 550 in the US (mean age was 19.52 in China and 19.29 in the US). The results indicated that partial measurement invariance of the SSOMI scale across China and the United States participants was established. Furthermore, we observed validity evidence for the SSOMI scale through its correlations with a well-established self-stigma measure and measures of depression, anxiety, and stress. Finally, we found that the SSOMI scale is more strongly linked to symptoms of depression, anxiety, and stress in China than it is in the United States, supporting previous research. These findings enable researchers to utilize the scale cross-culturally (i.e., with participants of Chinese and US origin), and to develop and implement interventions targeting mental illness stigma in both China and the United States.

## 1. Introduction

Stigma associated with mental illness has been observed around the world [1]. Furthermore, those experiencing mental illness stigmatization have reported a variety of negative outcomes, including lowered self-esteem, increased demoralization, impaired work performance, feelings of inadequacy, and elevated depression [2,3,4,5]. Despite its global impact, much of the research on the negative impacts of mental illness stigma has been conducted in the United States, and more is needed to assess cross-cultural differences [6,7]. An important prerequisite to examining mental illness stigma across cultures is to identify measures that assess this construct without systematic measurement bias. This study addresses this need by assessing the measurement equivalence (invariance) of the Self-Stigma of Mental Illness scale (SSOMI) [5] across China and the United States. Establishing invariance across China and United States is particularly important given that these countries are believed to illustrate two distinct cultural traditions (i.e., collectivism vs. individualism [8] that could impact how stigma is conceptualized [9]. Additionally, this study provides evidence of the construct validity of the SSOMI by examining its link to another form of stigma and psychological distress.

### 1.1. Mental Health Stigma

Researchers have identified different forms of mental health stigma, including public stigma and self-stigma [10], as well as the self-stigma of mental illness and the self-stigma of seeking psychological help [5]. Public stigma refers to societally held biases and prejudice against a group of individuals, while self-stigma refers to an individual’s internalization of these publicly held biases and prejudice. It is hypothesized that those who internalize the publicly held stereotypes into self-sigma will suffer negative consequences such as demoralization and reduced self-esteem and self-efficacy, as well as lower income and impaired work performance [11,12,13]. While both public stigma and self-stigma are linked to negative outcomes, research has shown that self-stigma is more closely related to constructs like low self-esteem because it describes an individual’s internalization of stereotypes, rather than a broader awareness of societal views [10,14].

In addition to delineating between public and self-stigma, researchers have identified two related, but distinct, forms of self-stigma: mental illness self-stigma and help-seeking self-stigma [5]. Mental illness self-stigma refers to an individual’s perception that having a mental illness would reduce one’s self-worth, while help-seeking self-stigma refers to an individual’s perception that seeking out psychological services would reduce one’s self-worth. Importantly, both forms of stigma are related to more negative psychological outcomes, including more negative perceptions of seeking out psychological services, and higher levels of shame, self-blame, and social inadequacy [5]. While help-seeking self-stigma is more closely linked to help-seeking outcomes [15], mental illness self-stigma may be a more salient predictor of psychological distress given its focus on the experience of mental illness.

### 1.2. Mental Illness Stigma and Cultural Identities

Mental illness stigma exists around the world, but it may present differently across cultures [16]. For example, in more collectivistic cultures like in China, Korea, and Latinx cultures, mental health stigma may be diffused through social structures that include families, friends, and other support systems [17,18]. This diffusion might help individuals who have mental illnesses dampen the extent to which they experience stigma [19]. In line with this, Chong and Wong [6] found that some Chinese people with mental illnesses in Hong Kong felt supported by others in their social networks. This may be different from the experience of stigma for individuals in individualistic cultures, like in the United States, where an individual may feel more pressure to address any mental illnesses by themselves because independence and self-sufficiency are often valued [20].

Yet, other researchers have found contrasting results. One study assessed stigma across participants in China and Canada, finding that Chinese participants reported a higher level of perceived stigma than the Canadian participants [7]. In another cross-cultural study that compared the stigma of depression among Chinese American and Caucasian American participants, Chinese participants reported higher levels of depression stigma than Caucasian American participants [21]. To explain this, the authors postulated that it may be more socially acceptable or normative to stigmatize mental illness for Chinese Americans relative to Caucasian Americans or that there is a larger amount of ‘distorted’ beliefs about depression in Chinese American culture [21].

### 1.3. Measuring the Self-Stigma of Mental Illness

The Self-Stigma of Mental Illness scale (SSOMI) was developed to assess mental illness self-stigma in the United States [5]. Adapted from the Self-stigma of Seeking Help scale (SSOSH) [22], the SSOMI is a 10-item scale that examines “the reduction in self-esteem and self-efficacy that results from receiving the label of mental illness” [5]. Another measure of mental illness self-stigma, the Internalized Stigma of Mental Illness Scale (ISMI) [23], has been widely used across many countries, including in China [24]. However, while the ISMI is an important stigma scale, there are some advantages of the SSOMI that warrant its examination across cultures. For example, the SSOMI (10 items) is a briefer measure than the ISMI (29 items), making the SSOMI less burdensome on participants. Additionally, the SSOMI asks participants about the anticipated experience of having a mental illness and how it would impact their self-esteem and self-worth, making it applicable to all participants, regardless of whether they currently have a mental illness. In contrast, many items on the ISMI assess views of others with a mental illness (e.g., “People with mental illness cannot live a good, rewarding life”) or only apply to individuals currently experiencing a mental illness (e.g., “People ignore me or take me less seriously just because I have a mental illness”). Finally, previous research has reported that some of the ISMI subscales may not demonstrate adequate internal consistency across all studies [24]. As such, it will be beneficial to further examine the SSOMI across China and the Unites States.

Given that measures cannot be assumed to validly assess their target construct in new populations, an important first step to utilizing the SSOMI across cultures is to examine the scales of psychometric properties using cross-cultural samples. For example, if group differences were to be observed when using the SSOMI across cultures, it would not be clear if this was due to a true difference between cultures or to measurement bias across cultures (e.g., different measurement error). One way to assess for measurement bias is to examine measurement invariance/equivalence (ME/I) across cultures [25].

Typically, three types of invariances are examined when assessing ME/I across groups: configural, metric, and scalar invariance. Configural invariance tests whether the factor structure is equivalent across different groups [25]. In the case of the SSOMI, researchers have used this as a single-factor scale, generating a total score for use in subsequent analyses [5]. If configural invariance is established, metric invariance is tested to assess whether the factor loadings for each item are the same across groups [25]. Establishing this form of invariance is necessary should a researcher be interested in comparing the strengths of the relationships between the measure and the related constructs across groups (e.g., comparing the relative strengths of correlations). Finally, if metric invariance is established, scalar invariance is tested to examine whether each item’s intercept/mean is equivalent across groups [25]. Establishing scalar invariance is believed to be necessary if group mean differences are being examined for the measure [26]. To our knowledge, the SSOMI has not been validated outside of the United States, reducing its utility in measuring self-stigma related to mental illness across cultures. Therefore, the current study assesses the invariance of the scale across the United States and China to determine whether it could be used in future cross-cultural research.

### 1.4. Present Study

In this study, we seek to: (1) establish the measurement invariance of the SSOMI across Chinese and US samples; (2) identify whether individuals from China (a collectivistic culture) endorse a higher level of self-stigma than those in the United States (an individualistic culture); and c) test two types of construct validity of the SSOMI in China and the United States. Specifically, convergent validity will be assessed by examining if the SSOMI is positively correlated with help-seeking stigma and predictive validity will be assessed by examining whether the SSOMI is related to distress levels (i.e., depression, anxiety, and distress). Additionally, the predictive validity of the SSOMI will be compared across Chinese and US samples to determine whether the SSOMI demonstrates differential associations with distress across the two countries.

## 2. Materials and Methods

### 2.1. Procedure and Participants

The current study has been approved by the research ethical board at Blinded For Review and the Institutional Review Boards at Blinded For Review. In China, convenience sampling was used to recruit 841 college students from a few departments at one university in Southern China. The survey link was sent to the individuals who monitor students’ academic progress, and they subsequently sent the link to students. Interested students then completed the survey at their convenience. In the United States, participants were recruited through introductory psychology and communication studies courses at a large midwestern university. The interested participants were directed to an online survey using Qualtrics software and completed the study in exchange for course credit. Participants had to be students at the university to participate. Prior to data collection, participants completed the consent form, which specified the purpose of the study, the potential benefits and risks of participation, and that participation would include the completion of a number of survey items about their beliefs about mental illness and seeking help, as well as items about their levels of distress. Participants were also informed that that participation was voluntary, and they could stop participation at any time during the data collection process. Initially, 841 participants in China and 626 in the United States agreed to participate and began the study. Of these, 487 participants in China and 550 in the U.S. completed the survey and correctly responded to the two attention check items. Data from participants who did not correctly respond to the attention check items or did not complete more than 50% of the survey were omitted from the analyses.

The majority of participants (mean age = 19.52 in China and 19.29 in the US) identified as women (69.6% women in China and 64.2% women in the US) and spanned across all years in school. Most participants in the Chinese sample were in their first year in college (48.3% first year, 27.1% second year, 10.3% third year, 5.3% fourth year, and 9.0% other), as were those in the US sample (49.3% first year, 27.1% second year, 10.0% third year, 11.5% fourth year, and 2.2% other). We also assessed previous help-seeking behavior, with 15.0% of Chinese participants and 34.7% of US participants reporting that they had sought help from a mental health professional within the previous five years. Participant race/ethnicity and sexual orientation were collected in the US sample, but not in the Chinese sample due to a lack of relevance and school administration restrictions. The majority of the US sample identified as European American (80.0%) and heterosexual/straight (86.0%).

### 2.2. Measures

Prior to data collection in China, all study measures were translated from English into Chinese, following the back-translation process [27].

The Self-Stigma of Mental Illness scale SSOMI) [5] is a 10-item scale measuring the self-stigma related to mental illnesses. Responses are rated on a 5-point Likert-type scale ranging from 1 = strongly disagree to 5 = strongly agree. Four items are reverse scored, and a composite score is calculated by averaging the ten items with higher scores, indicating higher levels of self-stigma. A sample item is “I would feel worse about myself if I had a mental illness.” [5] reported that the Cronbach’s alpha of their samples ranged from 0.91–0.92 in a sample of US college students and has yet to be used with a Chinese sample. In the current study, the SSOMI had a Cronbach’s alpha of 0.88 in the Chinese sample and 0.92 in the US sample. An alpha of above 0.65 is often considered acceptable in research involving human participants [28].

The Self-Stigma of Seeking Help scale (SSOSH) [22] is a 10-item scale measuring the self-stigma related to seeking psychological help. Responses are rated on a 5-point Likert-type scale ranging from 1 = strongly disagree to 5 = strongly agree. Five items are reverse scored, and a composite score is calculated by averaging the ten items with higher scores, indicating higher levels of self-stigma. A sample item is, “It would make me feel inferior to ask a therapist for help.” Previous research has shown that the SSOSH demonstrates internal consistency across international samples, ranging from 0.77–0.89 [14]. In this study, the SSOSH had a Cronbach’s alpha of 0.76 in the Chinese sample and 0.88 in the US sample.

The Depression Anxiety Stress Scale-21 (DASS [29] is a 21-item scale measuring depression, anxiety, and distress symptoms experienced over the previous week. Responses are rated on a 5-point Likert-type scale ranging from 0 = strongly disagree to 4 = strongly agree. Three seven-item subscales are calculated (i.e., depression, anxiety, stress) by averaging the seven items. A sample depression item is “I felt down-hearted and blue,” a sample anxiety item is “I felt I was close to panic,” and a sample stress item is “I found it hard to wind down.” Lovibond and Lovibond [29] reported subscale alphas ranging from 0.81 to 0.91 in their initial study, and previous research in China had reported alphas ranging from 0.80–0.83 [30]. In the current study, the Cronbach’s alpha ranged across the three subscales from 0.85 to 0.91 in the Chinese sample and from 0.87 to 0.91 in the US sample (Table 1).

### 2.3. Analytic Plan

To examine the ME/I of the SSOMI between the Chinese and United States samples, configural, metric, and scalar invariance were assessed using the sequential constraint approach [31] in MPLUS 6.11 with the MLR estimator [32]. The sequential constraint approach calls for the use of a series of increasingly constrained, multiple-group confirmatory factor analyses creating nested models which are then compared [25,33]. Configural invariance was assessed first (i.e., factor structure equivalence across samples), then metric invariance (i.e., item loading equivalence across samples), and finally scalar invariance (i.e., item intercept mean equivalence across samples [25]). Goodness of fit was assessed using the Comparative Fit Index (CFI > 0.95), Tucker Lewis Index (TLI > 0.90), Root Mean Square Error of Approximation index (RMSEA < 0.08), and the Standardized Root Mean Square Residual index (SRMR < 0.08), and the suggested cutoff of ΔCFI < −0.01 was used to determine whether two nested models were invariant [34]. If the nested model in which equality constraints were imposed did not produce a significant worse fit compared to the model in which no equality constraints were imposed, then invariance was established. Finally, we examined whether the SSOMI scale latent mean is significantly different between the Chinese and US samples.

After establishing measurement invariance, we assessed the validity of the SSOMI across China and the United States. First, we assessed the convergent validity (a type of construct validity) of the SSOMI across the two samples by examining the correlation between the SSOMI and SSOSH. These two constructs are theoretically linked and have been found to correlate with one another in previous research [5]. Next, we assessed the predictive validity (a type of construct validity) of the SSOMI across China and the United States by examining whether the SSOMI significantly predicted measures of depression, anxiety, and stress. If the SSOMI predicted these distress measures, we also examined whether there were differences in the strength of these relationships across the two countries by using moderation analyses.

## 3. Results

### 3.1. Descriptive Statistics

The scale means and standard deviations are reported in Table 1, and item level means, variance, skewness, and kurtosis are reported in Table 2. Group mean differences were assessed using SPSS v 28. There were no significant differences across the Chinese and US samples for anxiety (*t* = 1.57, *p* = 0.12), nor for help-seeking self-stigma (*t* = −1.51, *p* = 0.13), though there were significant group differences for depression (*t* = 4.18, *p* < 0.001, Cohen’s *d* = 0.26), stress (*t* = 3.78, *p* < 0.001, Cohen’s *d* = 0.24), and mental illness self-stigma (*t* = 6.50, *p* < 0.001, Cohen’s *d* = 0.40). Specifically, participants in the US reported higher levels of depression, stress, and mental illness self-stigma than participants in China.

### 3.2. Measurement Invariance

First, configural invariance was examined for the US and Chinese samples separately, loading the 10 items of the SSOMI onto a single latent factor. The model demonstrated poor fit in both the American (S-B χ^2^ (35) = 236.22; CFI = 0.91; SRMR = 0.05) and Chinese (S-B χ^2^ (35) = 204.89; CFI = 0.87; SRMR = 0.07) samples. Based on this poor fit, we reviewed the modification indices and a pattern emerged for both countries in which adding correlations between the negatively worded items would improve model fit. This suggested that method factors should be added to the model (one for the positively worded items and one for the negatively worded items). This is consistent with other studies that have found the need to add method factors to account for negatively worded items [35]. We added these method factors by creating two new latent variables in the model (positive and negative), and then loading the four reverse coded words onto the negative factor and the remaining six items onto the positive factor. We then re-ran the configural models across both samples. The resulting configural models fit the data in both the US (S-B χ^2^ (25) = 84.42; CFI = 0.97; SRMR = 0.03) and Chinese (S-B χ^2^ (25) = 66.18; CFI = 0.97; SRMR = 0.03) samples.

Using the model with method factors, we conducted a series of models to assess the configural, metric, and scalar invariance across the two samples. Table 3 summarizes these results. The SSOMI demonstrated full configural invariance across the two countries, so metric invariance was tested. The full metric invariance model (i.e., all item loadings constrained to be equivalent across groups) demonstrated a ΔCFI greater than the 0.01 cutoff [34], so we examined the modification indices to identify item loadings that should be freed. Modification indices indicated that freeing the item loading from item four to the positive method factor would improve fit, so we re-ran the model with this item loading freed across groups. The resulting model was a better fit, but still demonstrated a ΔCFI greater than 0.01 relative to the configural model. Modification indices were again consulted, and they indicated that the item loading from item five to the SSOMI latent factor should be free to improve model fit, and the resulting model (i.e., partial metric model) demonstrated a ΔCFI of less than 0.01 relative to the configural model.

The partial metric model was used to assess scalar invariance. To do this, constraints were imposed on the model, setting the item intercepts to be equivalent across countries. The full scalar model demonstrated a ΔCFI greater than 0.01 relative to the partial metric model, so we again went through the process of examining modification indices. After freeing the item intercepts for items three and six, the partial scalar model demonstrated a ΔCFI lower than 0.01 relative to the partial metric model. Overall, the results indicated that the SSOMI demonstrated full configural invariance and partial metric and scalar invariance across the American and Chinese samples. Finally, using the partial scalar invariance model, we compared the latent means between the Chinese and US samples. The results indicated that the SSOMI latent mean score for the Chinese participants was lower than that of the US participants (mean difference = −1.51, *p* < 0.01).

### 3.3. Construct Validity

After establishing partial measurement invariance, we examined the construct validity of the SSOMI. In both China and the United States, we correlated the SSOMI with the Self-Stigma of Seeking Help scale (SSOSH) to assess construct validity and the three subscales of the Depression Anxiety Stress Scale (DASS) to assess construct validity. The SSOMI was correlated with the SSOSH in both countries, providing evidence of construct validity (Table 4). In the United States sample, the SSOMI was significantly correlated with depression and stress, while the SSOMI was correlated with all three distress measures in the Chinese sample (Table 4).

Based on the significant correlations between the SSOMI and distress measures across China and the United States, we assessed whether the SSOMI had a significantly stronger relationship with the three distress measures in the Chinese sample relative to the US sample by conducting three moderation analyses using the PROCESS macro [36]. Using ‘Model 1′ in PROCESS, the standardized SSOMI was included as the predictor variable, a dummy coded country variable was included as the moderator (i.e., 0 = USA, 1 = China), and one of the standardized DASS subscales was used as the outcome. For the depression outcome, the regression was significant (F = 20.80, *p* < 0.001, R^2^ = 0.06). Both the SSOMI (B = 0.14, *p* < 0.001) and country (B = −0.17, *p* < 0.01) variables were significant predictors of depression. The interaction term was also significant (B = 0.14, *p* < 0.05), indicating that the relationship between the SSOMI and depression was moderated by country. Conditional effects analyses showed that the relationship between the SSOMI and depression was stronger in China (B = 0.28, *p* < 0.001) than in the US (B = 0.14, *p* < 0.001).

For the anxiety outcome, the regression was significant (F = 11.74, p < 0.001, R^2^ = 0.03). Neither the SSOMI (B = 0.07, *p* = 0.06) nor country (B = −0.03, *p* = 0.66) variables were significant predictors of anxiety, but the interaction term was significant (B = 0.20, *p* < 0.01), indicating that the relationship between the SSOMI and anxiety was moderated by country. Conditional effects analyses showed that the relationship between the SSOMI and anxiety was stronger in China (B = 0.27, *p* < 0.001) than in the US (B = 0.07, *p* = 0.06).

Finally, for the stress outcome, the regression was significant (F = 18.54, *p* < 0.001, R^2^ = 0.05). Both the SSOMI (B = 0.09, *p* < 0.05) and country (B = −0.15, *p* < 0.05) variables were significant predictors of stress. The interaction term was also significant (B = 0.21, *p* < 0.01), indicating that the relationship between the SSOMI and stress was moderated by country. Conditional effects analyses showed that the relationship between the SSOMI and depression was stronger in China (B = 0.30, *p* < 0.001) than in the US (B = 0.09, *p* < 0.05).

## 4. Discussion

This study assessed the measurement invariance of the SSOMI across China and the United States, with results showing full configural invariance after the addition of method factors and partial metric and scalar invariance. Additionally, this study found that the SSOMI demonstrated construct validity in both the Chinese and US samples, though there were differences in the strengths of some of the relationships across the two countries. Ultimately, these results support the use of the SSOMI to assess cross-cultural differences related to mental illness self-stigma across China and the United States.

Full configural invariance was found for the SSOMI between China and the United States, indicating that the factor structure of the SSOMI is equivalent across these two countries. This suggests that the self-stigma of mental illness construct exists in both Chinese and US culture. Importantly, configural invariance was only established after the addition of method factors for the SSOMI. This is a novel finding given that previous research found that a unidimensional scale was the best fit [5], and it may be due to the fact that we examined the SSOMI independently, while [5] and colleagues examined a CFA model with items from both the SSOMI and SSOSH together. Further, other researchers have noted similar findings, in that adding a method factor to the general factor could improve the model fit of the data [35,37]. Future research is needed to verify the existence of the method factor across additional samples.

While full configural invariance was established, partial metric invariance was observed. Specifically, all item factor loadings were equivalent across the Chinese and US samples except for items four (loading on the positive method factor) and five (loading on the SSOMI general factor). The loading of item four (“My self-esteem would decrease if I had a mental illness”) on the positive method factor was higher in the Chinese sample than in the US sample. Given that this was a loading on a method factor, it does not indicate that the item has a differential link to the latent SSOMI construct across the two countries, but it does suggest that it is more strongly linked to the positively worded method effect in China. One possibility is that ‘self-esteem loss’ holds a stronger valence in China than in the United States. This is consistent with previous research that found Chinese individuals reported lower self-esteem than those in the United States, and the difference was largely driven by ‘negatively’ worded self-esteem items [38]. Future research is needed to replicate this finding to ensure it is not specific to this sample. However, item five (“My view of myself would not change just because I had a mental illness”), which is reverse coded, had a stronger loading on the general factor in the US sample than in the Chinese sample, indicating that this item may be a stronger indicator of mental illness self-stigma in the United States than in China. It is possible that mental illness self-stigma may be more strongly related to self-acceptance in the United States than in China. This makes sense given that the SSOMI scale was developed and validated using a U.S. sample [5], and its items are likely more characteristic of an individualistic perspective on the effects of mental illness (i.e., more strongly linked to how an individual views themselves).

Partial scalar invariance was also observed across the two samples with only the intercepts for items three (“Having a mental illness would make me feel less intelligent”) and six (“It would make me feel inferior to have a mental illness”) showing non-invariance. In both cases, the intercept was larger in the US sample, meaning that US participants tended to report that having a mental illness would make them feel less intelligent and more inferior more than those in the Chinese sample. For item three, one possibility is that Chinese individuals may not believe mental illness has as strong an impact on intelligence as their American counterparts do because they believe they have more ‘control’ of their intelligence outside of the influence of mental illness. For example, previous research suggests that Chinese individuals are more likely to view intelligence as something achieved through effort rather than as a result of fixed ability [39]. The non-scalar-invariance of item six may be due to the cultural differences between China and the Unites States. Specifically, those in the United States are more likely to identify as having an individualistic cultural background, and thus may believe that mental illness will more strongly impact their self-worth. However, scholars have recently called for more research on this topic given that research is only beginning to uncover the role of culture in the internalization of stigma [40].

Statistically, it is logical to freely estimate parameters in order to improve model fit in general, for more parameters will help the estimated model be closer to the observed data [41]. There are salient differences between individuals from individualistic cultures and collectivistic cultures in terms of how the self is construed. Independent self-construal characterizes people from individualistic cultures and their behaviors are often determined by their own thoughts, feelings, and other personal attributes. On the contrary, the interdependent construal of the self represents people from collectivistic cultures and these individuals’ behaviors are usually determined in reference to others’ expectations and social norms [42]. Previous findings indicate that an interdependent self-construal (in this case, the Chinese participants) essentially serves a social support system [17] that may help diffuse the self-stigma of mental illness [19]. As such, individuals from cultures characterized by independent self-construal (in this case, the US participants) may experience lower self-worth as their level of mental illness self-stigma increases, since such a diffusion process does not exist.

Though two items from the SSOMI demonstrated non-metric invariance and two items demonstrated non-scalar invariance, the results still support the use of the SSOMI in cross-cultural research across China and the United States. Specifically, 90% of the items demonstrated metric invariance on the general SSOMI factor (item four′s loading was non-invariant on the positive method factor), while 80% of items demonstrated scalar invariance. This level of invariance is similar to, if not better than, other stigma measures that have been examined across cultures [43], and it allowed us to examine the construct validity of the SSOMI across China and the United States. First, the SSOMI demonstrated construct (convergent) validity through its significant relationship with the SSOSH in both China and the United States. Previous research has found these two measures to be correlated in US samples [5], but this is the first study showing that this relationship is present in China.

Overall participants from the United States reported a higher SSOMI score than participants from China. It is possible that this finding is because self-stigma is conceptually more congruent with the individualistic American culture [5]. However, this finding needs to be replicated before broad conclusions can be made given than this is the first Chinese sample to utilize the SSOMI.

The SSOMI was also correlated with the three subscales of the DASS (i.e., depression, anxiety, and stress) to examine construct (predictive) validity across the two countries. Interestingly, the SSOMI was significantly related to each of the three distress scales in the Chinese sample but was only significantly related to depression and stress in the US sample. Furthermore, the correlations were noticeably stronger in the Chinese sample, and moderation analyses confirmed that the relationships between the SSOMI and the DASS subscales were significantly stronger for the Chinese participants than for the US participants. These findings are consistent with previous research that found self-stigma had a larger impact on distress in Asian populations [44,45,46]. Future researchers might build on these findings by examining the longitudinal relationships between mental illness self-stigma and distress to better understand the directionality of these relationships. For example, increases in self-stigma could result in higher levels of distress due to the experience of shame that is common with stereotyped or prejudiced messages. Alternatively, increased distress could subsequently result in increased self-stigma due to the increased personal relevance of publicly held stereotypes.

Regarding the lack of significant correlation between anxiety and SSOMI among the US sample, we postulated the following reason: according to a recent study, it is possible that anxiety has become a “normalized” mental health condition for US folks, and that it is no longer significantly related to mental health stigma [2]. Although this could potentially explain our current finding, further research is needed to provide more evidence.

### Limitations and Implications for Future Research

This study had a number of limitations. First, due to the homogeneity of the current sample, the current study cannot claim that the observed moderating effect of country of origin can be generalizable to other populations of interest. Future research needs to consider conducting similar research including more diverse populations in order to draw more valid conclusions regarding the relationships between variables. In addition, not only were our samples homogenous in terms of nationality, but our samples also only included college students. Future studies need to include both younger and older participants to expand the external validity of the SSOMI scale. Second, our sample only included participants from China and the U.S., which is far from generalizable to other parts of the globe. We believe that future research could include participants from more diverse cultural backgrounds and further test the relationship, so that more evidence can be gathered regarding the impact of country of origin on SSOMI.

Another important aspect for future study is to consider other ways to address the poor measurement fit due to reverse worded items. We chose to add method factors in the current paper, which helped improve the model fit. However, with the confounding influence of culture and wording of items, it has been pointed out that adding a method factor for these reverse worded items could often contaminate the factor structure, and therefore the “expanded format” is recommended [47]. The “expanded format” instructs researchers to replace the original item by a full sentence through inserting all relevant Likert scale frequency words. The participants are forced to be attentive to the nuanced difference between each statement of the item, thus reducing potential carelessness and confusion [47].

In addition, the potential impact from the ebb and flow of the semester cannot be ignored. Specifically, our Chinese sample might have been affected more by the exams toward the end of the semester, hence experiencing a higher level of anxiety, while their US counterparts did not. This could potentially contribute to the mean difference of the anxiety level across Chinese and US samples. Future studies need to consider this when recruiting participants.

Despite these limitations, the results highlight important areas for future research. For example, future studies are needed to assess how stigma differentially impacts help-seeking behaviors and other related outcomes across different cultures. Specifically, stigma is a “context based” and “multifaceted” construct [48] suggesting that different contexts (e.g., cultures or countries) could impact how it influences behavioral decision making. Future research might build on our findings by exploring how cultural identities interact with stigma to predict help-seeking behaviors. Additional research might assess factors that predict stigma across cultures. One example is socialized gender role norms, which have been identified as an important predictor of stigma in the United States [49]. Less research has focused on how gender role norms predict stigma across cultures.

These results also have implications for clinicians. First, these results support the notion that the self-stigma of mental illness is present in individuals across China and the United States. Second, the results showed that mental illness self-stigma is linked to distress outcomes across both countries. This suggests that clinicians may benefit from attending to issues of self-stigma with distressed clients given that a client’s symptom presentation in the counseling process may be related to their experience of self-stigma. One option is that clinicians could hold an open and empathetic perspective and invite clients to discuss their experience of mental illness self-stigma in an effort to work through their stigma concerns.

## 5. Conclusions

Applying measurement invariance models to the current data recruited from China and the United States, this study provided evidence for partial measurement invariance of the SSOMI scale across China and the United States, enabling researchers to utilize the scale cross-culturally (i.e., with participants of Chinese and US origin). Furthermore, we observed construct validity evidence for the SSOMI through its correlations with a well-established self-stigma measure, the SSOSH, and measures of depression, anxiety, and stress. Finally, this study found that the SSOMI is more strongly linked to symptoms of depression, anxiety, and stress in China than in the United States, supporting previous research, which is an important finding for researchers who are developing and implementing interventions targeting mental illness stigma in both China and the United States.

## Figures and Tables

**Table 1 ijerph-19-02344-t001:** Means, standard deviations, and Cronbach’s alpha for all variables, split by sample.

Variable	Sample					
	China (*n* = 487)			United States (*n* = 550)		
	Mean	SD	α	Mean	SD	α
SSOMI	2.65	0.68	0.88	2.95	0.81	0.92
SSOSH	2.45	0.53	0.76	2.39	0.74	0.88
Depression	0.64	0.62	0.89	0.81	0.74	0.91
Anxiety	0.64	0.58	0.85	0.70	0.70	0.87
Stress	0.84	0.72	0.91	1.01	0.73	0.87

Note. SSOMI = Self-Stigma of Mental Illness scale; SSOSH = Self-Stigma of Seeking Help scale; SD = standard deviation; α = Cronbach’s alpha.

**Table 2 ijerph-19-02344-t002:** SSOMI item, mean, variance, skewness, and kurtosis across China and the United States.

Item	Country	Mean	Variance	Skewness	Kurtosis
SSOMI 1	China	2.97	1.18	−0.30	−1.00
	USA	2.64	1.22	0.09	−1.03
SSOMI 2	China	3.02	0.99	−0.22	−0.55
	USA	3.52	1.15	−0.38	−0.78
SSOMI 3	China	2.54	1.00	−0.53	0.68
	USA	2.58	1.22	0.18	−1.04
SSOMI 4	China	2.78	1.10	0.02	−0.80
	USA	3.19	1.12	−0.46	−0.71
SSOMI 5	China	2.53	0.85	0.40	−0.22
	USA	3.21	1.06	−0.27	−0.73
SSOMI 6	China	2.20	0.87	0.68	0.12
	USA	2.72	1.17	0.12	−0.90
SSOMI 7	China	2.15	0.67	0.73	0.96
	USA	2.78	0.93	0.29	−0.39
SSOMI 8	China	3.01	1.07	−0.25	−0.83
	USA	2.91	1.16	−0.15	−0.98
SSOMI 9	China	2.84	0.84	−0.09	−0.08
	USA	3.27	0.96	−0.41	−0.47
SSOMI 10	China	2.45	0.97	0.30	−0.49
	USA	2.95	1.18	−0.21	−0.97

*Note.* SSOMI = Self-Stigma of Mental Illness.

**Table 3 ijerph-19-02344-t003:** Measurement invariance of the SSOMI across countries (*n*= 1036).

	S-B χ^2^	*df*	SRMR	RMSEA	RMSEA CI	TLI	CFI	ΔCFI	Model Comparison
Configural	150.56	50	0.028	0.062	0.051–0.074	0.949	0.970	--	
Metric									
Full	243.59	67	0.050	0.071	0.062–0.081	0.933	0.950	−0.020	Configural
Partial	196.54	65	0.048	0.063	0.053–0.073	0.948	0.963	−0.007	Configural
Scalar									
Full	255.63	72	0.043	0.070	0.061–0.080	0.935	0.948	−0.015	Partial Metric
Partial	228.99	70	0.040	0.066	0.057–0.076	0.942	0.955	−0.008	Partial Metric

*Note.* S-B χ^2^ = Satorra–Bentler scaled chi-square values; *df* = degrees of freedom; SRMR = Standardized Root Mean Square Residual; RMSEA = Root Mean Square Error of Approximation; RMSEA CI = 90% confidence interval for the RMSEA; TLI = Tucker Lewis Index; CFI = Comparative Fit Index. The configural model results refer to the configural model in which the Chinese and US samples were modeled conjointly.

**Table 4 ijerph-19-02344-t004:** Zero-order correlations between the SSOMI and related constructs in China and the United States.

	SSOMI	SSOSH	Dep	Anx	Stress
SSOMI	--	0.62 ***	0.33 ***	0.27 ***	0.27 ***
SSOSH	0.58 **	--	0.33 ***	0.28 ***	0.28 ***
Dep	0.14 **	0.07	--	0.79 ***	0.77 ***
Anx	0.07	0.02	0.66 ***	--	0.83 ***
Stress	0.10 *	−0.01	0.73 ***	0.75 ***	--

Note. Chinese correlations are above the diagonal and US correlations are below the diagonal. SSOMI = Self-Stigma of Mental Illness scale; SSOSH = Self-Stigma of Seeking Help scale; Dep = Depression; Anx = Anxiety. * *p* < 0.05; ** *p* < 0.01; *** *p* < 0.001.

## Data Availability

The data presented in this study are available upon request from the corresponding author.

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
