# Peer review of "Measurement Invariance of the Self-Stigma of Mental Illness Scale: A Cross-Cultural Study"

_ijerph, 2022, doi:10.3390/ijerph19042344_

Round 1
Reviewer 1 Report
Measurement Invariance of the Self-Stigma of Mental Illness 2 Scale: A Cross-cultural Study
The study aimed to assess the measurement invariance of the SSOMI across China and the US. This is a well written manuscript and I congratulate the authors on the job well-done.
comments
There are some grammatical errors and tense inconsistencies which need attention.
The authors talk of validity, but donot specify what type of validity is being referred to. Validity is large construct; I suggest that the authors narrow down to the relevant type of validity.
It would be good for the authors to provide some details about the methods, , how were they recruited and approached, what were the inclusion/exclusion criteria, how was the sampling done?
A comment on this statement – Based on this poor fit, we reviewed 246 the modification indices and a pattern emerged for both countries in which adding cor- 247 relations between the negatively worded items would improve model fit. I think negatively worded items tend to split into a factor of their own ( check literature by Xijuan Zhang et al 2016).
It would be good to make a subject matter sense of why freeing item 4 results into a better model.
Same for items three and six for the partial scalar model.
In table 4, it is difficult to differentiate the Chinese from the USA correlations, could the authors please improve on this visibility?
Reviewer 2 Report
1) Literature review : Please provide more recent studies related to mental illness stigma and cultural identities.
2) Check for spelling error (page 3, para 1)
3) Please provide reference citation for acceptable cut-off value of Cronbach's alpha that the author refer for all measures (page 4, para 3)
4) Sample: Explain participants in more details i.e the author did explain participants are from all years in school but what type of school and justify the reason for selecting the participants.
5) Define sampling technique used in this study.
Reviewer 3 Report
The present research investigated measurement invariance of the Self-stigma of Mental Illness scale (SSOMI) across Chinese and US samples. The questionnaire was administered to the two groups of participants and the responses were assessed in terms of configural, metric, and scalar invariance, using the sequential constraint models. The participants also answered the Self-Stigma of Seeking Help Scale (SSOSH) and Depression Anxiety Stress Scale-21 (DASS). The SSOSH was used to confirm the validity of SSOMI, and the DASS was used to examine whether SSOMI differentially relates to those mental illnesses across Chinese and US participants. Overall, it was concluded that the SSOMI is suited to be used cross-culturally. The mental illnesses measured with the DASS were found to be related to self-stigma, but the relationship was stronger for the Chinese than for the Americans, suggesting that culture influences how mental illness impacts self and how people experience self-stigma.
The paper is well written, and the investigated issue is interesting as well as crucial for the development of mental health treatment. The research method and analyses used were appropriate. The results were well presented and thoroughly discussed. However, the manuscript lacks explanations on some important issues. Also, the discussion can be deepened by focusing more on how self-stigma and mental illness relates to individuals’ self-construal, which is often influenced by culture. My concerns are as below.
Major points
1) In the introduction, the study purpose c), stated on page 3 line 137, is not well explained. How and why SSOSH and DASS are used, and what can be expected for the results should be described. Although they are briefly explained in the first paragraph of page 6, more detailed information should be included in the introduction.
2) Information about participants in the procedure and participants section is insufficient. Who are they and how were they recruited? Also, was the study online or not? If it was a pencil and paper format, please describe the environment your participants were in when they answered it (in a lecture room, for example).
3) In the discussion, a number of freed items from the model were described and you have attempted to attribute the group differences to the US and Chinese cultural tendencies. Although I have enjoyed your argument, I think such argument should be expanded to more general findings of the present research rather than focusing on each questionnaire item. Since this research is about self-stigma, I would like to see more discussions about the relationship between how people understand self and stigma. Individualistic and collectivistic cultural tendencies in behaviors and thinking styles stem from how individuals understand self (self-construal, see Markus & Kitayama (1991). Culture and the self: Implications for cognition, emotion, and motivation. Psychol. Rev. 98, 224-253). So, I think it is important to describe how mental illness affect people’s self-construal in general, then further discuss how individualistic/collectivistic tendencies lead them to perceive self in particular way and how it connects to experiencing stigma.
4) In the paragraph starting from page 10 line 399, you suggest that the SSOMI performs well cross-culturally. It would be nice if you also suggest how the culturally variable items should be treated in the future. Is removing them from the analysis recommended? Can they be used to highlight some cultural differences?
5) In the discussion section, the result that overall SSOMI was higher for the US than China is not discussed well. I understand that the results established that the relationship between mental illness and stigma is stronger for Chinese than Americans, but I think you should discuss more about the overall group differences in the mean scores of each measure found in the present results.
6) For the US participants, SSOMI was not correlated with anxiety while depression and stress were, though the relations were much weaker than in the Chinese data. Indeed, the US anxiety measure is only one mental illness found to be unrelated to self-stigma. Is there something special going on with how Americans experience anxiety? Please discuss more about this.
7) You have stated that your sample homogeneity is a limitation. I agree, but your sample is limited not only in terms of nationality but also in terms of age (very young adults). Please discuss the scope of generalizability of the current findings to different age groups.
Minor points
8) The United States is sometimes spelled out and sometimes written as the US. It is better to make it consistent throughout the manuscript.
9) Some paragraphs are indented unnecessarily.
10) Page 4 line 144, the sentence “Prior to data collection…” should be moved to the Measures.
11) Page 4 line 151, should “voluntartteny” be voluntary?
12) The Greek alphabet of delta for delta CFI is not appearing correctly in the manuscript in lines 213, 260, 265, 268.
13) Page 6 line 222, does “help-seeking self-stigma” in this sentence mean SSOSH? If so, please say so.
14) Should the font for statistical indices such as t and p be italicized?
15) On Table 4, the Chinese correlation between SSOSH and Depression says -.33. I think this is a mistake and it should be .33, but if it is correct, please explain the negative correlation thoroughly in the discussion.
16) Page 11 line 444, the number 5 in the end should be in the next line.
Reviewer 4 Report
The ‘Conclusions’ section is too laconic and should be expanded a bit.
The literature is relevant and up-to-date as well as the sources are rich; however, there is often no correlation between the sources in the references and the corresponding sources in the body of the text, i.e. it happens that the source mentioned in the body does not appear in the references and vice versa: the source in the references does not have its equivalent in the body of the paper. Besides, there are some technical or editorial errors or omissions and appropriate amendments and complements are required.
There are the following sources in the corpus of the paper but I can’t find them in the references:
– ‘Ritsher et al., 2003’ (line 93).
– ‘Cheung & Rensvold, 2002’ (line 128).
There are the following sources in the references but I can’t find them in the corpus:
– ‘Brown, T. A. (2015)’ (line 476).
– ‘Chang, M. X., Jetten, J., Cruwys, T., & Haslam, C. (2017)’ (line 477).
– ‘Conrad, M. M., & Pacquiao, D. F. (2005)’ (line 485).
– ‘Crabtree, J. W., Haslam, S. A., Postmes, T., & Haslam, C. (2010)’ (line 492).
– ‘Cruwys, T., & Gunaseelan, S. (2016)’ (line 494).
– ‘Henry, J. D., & Crawford, J. R. (2005)’ (line 502).
– ‘Ryder, A. G., & Chentsova-Dutton, Y. E. (2012)’ (line 537).
– ‘Wang, X., Huang, X., Jackson, T., & Chen, R. (2012)’ (line 561).
– ‘Wong, Y. J., Tran, K. K., Kim, S-H., Kerne, V. V. H., & Calfa, N. A. (2010)’ (line 565).
Other incorrect notations in the references:
– The source ‘Lannin et al.’ is marked as 2013 in the body of the paper (see line 62) but in references it is marked as 2015 (see line 512).
– The source ‘Vogel et al.’ is marked as 2006 in the body of the paper (see lines 90 and 177) but in references it is marked as 2007 (see line 559).
– The source ‘Yang’ is marked as 2008 in the body of the paper (see line 429) but in references it is marked as 2007 (see line 569).
– The source ‘Ryder, 2008’ in the body of the paper (see lines 30 and 79) should be written as ‘Ryder et al., 2008’.
Round 2
Reviewer 3 Report
I am satisfied with the corrected version, but there seem to be a quite few small typing mistakes such as extra spacing and periods/commas. I have highlighted possible mistakes on the attached. I recommend you to carefully go through them at the proofing process.
